# Supercapacitor Properties of rGO-TiO_2_ Nanocomposite in Two-component Acidic Electrolyte

**DOI:** 10.3390/ma15217856

**Published:** 2022-11-07

**Authors:** Yury M. Volfkovich, Alexey Y. Rychagov, Valentin E. Sosenkin, Sergey A. Baskakov, Eugene N. Kabachkov, Yury M. Shulga

**Affiliations:** 1A.N. Frumkin Institute of Physical Chemistry and Electrochemistry, Russian Academy of Sciences, Leninsky pr. 31, 119071 Moscow, Russia; 2Federal Research Center of Problem of Chemical Physics and Medicinal Chemistry, Russian Academy of Sciences, 142432 Chernogolovka, Russia; 3Institute of Solid State Physics, Russian Academy of Sciences, 142432 Chernogolovka, Russia; 4Department of functional polymer materials, National University of Science and Technology MISiS, Leninsky pr. 4, 119049 Moscow, Russia

**Keywords:** supercapacitor, rGO/TiO_2_ nanocomposite, aerogel, porous structure, hydrophilic–hydrophobic properties, electrochemical hydrogenation, pseudocapacitance

## Abstract

The electrochemical properties of the highly porous reduced graphene oxide/titanium dioxide (rGO/TiO_2_) nanocomposite were studied to estimate the possibility of using it as a supercapacitor electrode. Granular aerogel rGO/TiO_2_ was used as an initial material for the first time of manufacturing the electrode. For the aerogel synthesis, industrial TiO_2_ Hombikat UV100 with a high specific surface area and anatase structure was used, and the aerogel was carried out with hydrazine vapor. Porous structure and hydrophilic–hydrophobic properties of the nanocomposite were studied with a method of standard contact porosimetry. This is important for a supercapacitor containing an aqueous electrolyte. It was found that the hydrophilic specific surface area of the nanocomposite was approximately half of the total surface area. As a result of electrochemical hydrogenation in the region of zero potential according to the scale of a standard hydrogen electrode, a reversible Faraday reaction with high recharge rate (exchange currents) was observed. The characteristic charging time of the indicated Faraday reaction does not exceed several tens of seconds, which makes it possible to consider the use of this pseudocapacitance in the systems of fast energy storage such as hybrid supercapacitors. Sufficiently high limiting pseudo-capacitance (about 1200 C/g TiO_2_) of the reaction was obtained.

## 1. Introduction

According to the general accepted definition, which was given first by Conway, electrochemical supercapacitors (ECSCs) are devices where quasi-reversible electrochemical charging–discharging processes occur [1]. The shape of galvanostatic charging and discharging curves is close to linear. In other words, it is close to the shape of corresponding dependences for conventional electrostatic capacitors [1,2,3,4,5]. ECSCs are divided into double-layer capacitors (DLCs), pseudocapacitors (PsCs) and hybrid supercapacitors (HSCs). EDLCs, which are based on charging an electric double layer (EDL) of electrodes, contain electrodes based on highly-dispersed carbon materials (HDCM) with high specific surface area (SSA). The SSA values reach 500–2500 m^2^/g. Activated carbons (AC), carbide carbons, aerogels, xerogels, soots, nanotubes, nanofibers, graphenes, etc., are related to HDCM. Fast quasi-reversible electrochemical reactions occur in PsC electrodes, which are based on electrically-conducting polymers (ECP) (polyaniline, polythiophene, polypyrrole, etc.). Some oxides of metals possessing several oxidation degrees are also used.

Compared with accumulators, the ECSCs advantages are as follows: (i) higher power characteristics, (ii) higher cyclic resource (up to hundreds of thousands of cycles for high power supercapacitors) and (iii) reliable operation under the conditions of extreme temperatures (from −50 °C to 60 °C). Moreover, different types of ECSCs can be charged and discharged over a very wide range of time (from fractions of a second to hours). This expands the possible fields of their application: ECSCs are used in electric vehicles, cars, diesel locomotives (for starting internal combustion engines) as well as different electronic devices. Regarding electric vehicles, ECSCs can be used in combination with fuel cells for afterburner modes. ECSCs are divided into two main types: (i) power or pulse capacitors possessing high power density and (ii) energetic capacitors, which are characterized by high specific energy. Accordingly, each type of ECSC is suitable for one or other field of application [6,7,8].

The electrodes of industrial supercapacitors are most often made on AC. These supercapacitors possess very high stability under cycling (charging–discharging). However, their specific capacity, as a rule, is not high enough since only the EDL capacitance is involved [9,10,11,12].

Oxides of metals, for which different oxidation degrees are specific, are also used for the preparation of electrodes for PsCs. Such oxides as RuO_2_, NiO, Ni(OH)_2_, MnO_2_, Co_2_O_3_, IrO_2_, FeO, TiO_2_, SnO_2_, V_2_O_5_ and MoO_3_ are used (see reviews [13,14]). The insertion of these into an electrode mixture causes an increase in total capacitance due to the pseudocapacitance of faradaic reactions. Among oxides of transition metals, TiO_2_ possesses the following advantages: (i) sufficiently higher capacitance compared with other oxides. This material is also characterized by non-toxicity, cheapness, availability and durability. As a result, TiO_2_ is considered one of the most promising among pseudocapacitive materials. However, high capacitance values can be obtained only for nanosized TiO_2_ particles, which are inserted into a conductive matrix with enough open pores since this oxide is dielectric. This distinguishes the composites used for PsC electrodes from sorbents, where the active component is distributed throughout the volume of crystalline [15,16] or amorphous [17] titanium dioxide.

Use of a graphene-based carbon matrix with open pores is most reliable since reduced graphene oxide (rGO) is characterized by high SSA, and high conductivity was studied [18,19,20]. Graphene-like materials (GLMs) such as carbon nitride [21] or carbon nanoribbons [22] are used for the preparation of composites with wide functional purposes; in particular, composites of TiO_2_ with GLMs have already been tested as ECSC electrodes [23,24,25,26,27].

Electrochemical properties of nanocrystalline TiO_2_ in phases of anatase (TiO_2_-A), anatase/rutile (TiO_2_-C) and rutile (TiO_2_-R) has been reported [28]. Cyclic voltammetry (CV), galvanostatic charging–discharging and impedance spectroscopy were applied to the analysis of the electrochemical properties of electrodes. The results indicate very good electrochemical properties of the TiO_2_-A electrode due to its high SSA, developed porous structure and low resistance of charge transport. The CV study gave the capacitance of 362 F/g at a low scanning rate of 10 mV/s in a 0.5 M KCl solution. Moreover, galvanostatic charging–discharging gives specific capacitance of 312 F/g at this scanning rate. Based on high values of energy density (882 Wh/h) and power density (1260 kWh/kg), it has been concluded that the TiO_2_-A electrode is prospective material for supercapacitor preparation.

Comparing the values of the specific capacitance of TiO_2_ with corresponding magnitudes for carbon materials, it is necessary to take into consideration that the density of rutile (4.23 g/cm^3^) and anatase (3.78 g/cm^3^) is approximately two times higher than that of carbon (~2 g/cm^3^).

rGO/TiO_2_ nanocomposites were synthesized with a simple and eco-friendly method of microwave irradiation [23]. XRD analysis showed a tetragonal anatase phase of pure TiO_2_ and TiO_2_ in the nanocomposite. The sizes of crystallites were estimated as 2.8 nm and 2 nm respectively. SEM(Scanning Electron Microscope) and HR-TEM(High Resolution Transmission Electron Microscopy) observations showed spherical TiO_2_ nanoparticles on graphene sheets. The TGA data demonstrated higher thermal stability of rGO/TiO_2_ nanocomposites compared with grapheme oxide and TiO_2_ nanoparticles. The nanocomposites, which were used as supercapacitor electrodes, showed a higher value of specific capacitance (585 F/g) at 1 A/g in 1 M H_2_SO_4_ compared with rGO (174 F/g) and the TiO_2_ electrode (66 F/g). Improved capacitance characteristics are caused by intercalation of TiO_2_ nanoparticles on graphene sheets. The method of microwave irradiation provides viable, inexpensive and easy synthesis of various composites of rGO/metal oxide with promising properties for use as energy storage devices in supercapacitors for usage as energy storage devices.

rGO/TiO_2_ and MWCNTs/TiO_2_ (where MWCNTs are multiwalled carbon nanotubes) electrode materials were synthesized with a hydrothermal method [29]. The synergetic effect of the rGO/TiO_2_ electrode increases the specific capacitance up to 558 F/g at 1 A/g. Its cycle stability is 100% up to 5000 cycles. A fabricated symmetrical supercapacitor device showed the maximal energy density of 14 kWh/kg and power density of 9.6 kW/kg at 0.5 and 32 A/g, respectively; a 1 M H_2_SO_4_ solution was used. The electrode possesses very good cycle stability; this allows one to use it for energy storage devices.

The excellent performance of TiO_2_ (Hombikat UV100, Sachtleben Chemie GmbH, Duisburg, Germany) coatings has been suggested. It is much higher compared with coatings produced from P25 TiO_2_ under gas phase UV photocatalytic oxidation of two very different molecules with various behaviors in the flow of methyl ethyl ketone (MEK) and H_2_S [30]. According to the isotherms of N_2_ adsorption–desorption, the TiO_2_ phase was characterized by a high SSA of 330 ± 15 m^2^/g, and the SSA of micropores was about 243 m^2^/g.

The aim of this work was to obtain highly porous rGO/TiO_2_ nanocomposites and investigate their electrochemical characteristics as an electrode for supercapacitors. As opposed to previous works, the initial material for the electrode preparation was granulated rGO/TiO_2_ aerogel. In order to obtain this aerogel, the industrially produced TiO_2_ (Hombikat UV100) with high specific surface area and anatase structure was used. Graphene was reduced with hydrazine vapor, this approach allowed us to dope it with nitrogen. An important feature of this work is a detailed study of the porous structure and hydrophilic–hydrophobic properties of the rGO/TiO_2_ nanocomposite since the porous structure of carbon materials involves both hydrophilic and hydrophobic pores. It is important in our case since this nanocomposite was investigated in aqueous electrolyte, in H_3_PO_4_ and ZnBr_2_ solutions.

## 2. Experimental Section

### 2.1. Synthesis of rGO/TiO_2_ Composite

In order to obtain composite (70:30), 40 g of aqueous suspension of graphite oxide (1.3 mass %), 0.222 g TiO_2_ (Hombikat 100 UV) and 40 cm^3^ of twice deionized water were inserted into conical flask (100 cm^3^) and dispersed using a MEF 93.1 ultrasonic dispersant (LLC “MELFIZ”, Moscow, Russia) at 22 kHz and ultrasonic intensity of 250 W/cm^2^ during 30 min. In order to obtain composites with lower TiO_2_ content, its amount was reduced in the initial mixture. As supposed, ultrasonic treatment causes splitting of graphite oxide into single-layer (predominantly) and few-layer sheets of graphene oxide. For aerogel granules, a device consisting of glass Dewar vessel and solid copper cylinder was used. The cylinder (50 mm of a diameter and 250 mm of a length) was inserted into the Dewar vessel. A copper plate (150 mm × 150 mm × 15 mm) was installed on a cylinder. The device was supplied by a thermometer to control the platform temperature. The operation principle is to decrease the temperature of the platform down to −80 °C–−90 °C using liquid nitrogen in Dewar vessel. To obtain granules, the suspension was inserted to a copper plate in the form of drops using a dosing pipette. The granule size was controlled by the stroke of the pipette piston.

When the plate was filled with granules, they were removed with a putty knife, inserted into a freeze-drying container cooled below 0 °C for freeze drying. Frozen granules were loaded into an IlShin BioBase FD5512 freeze-dryer (Seoul, South Korea). The condenser volume was 12 L. The granules were kept at a pressure of 5 mTorr and a condenser temperature of −45 °C–−55 °C. The drying time was 72 h. Dry granules were treated with hydrazine vapor in hermetic polypropylene container at 55–60 °C during 48 h. During this reducing treatment, the color of granules was changed from light-grey to black (Figure 1).

### 2.2. Physical Techniques

A Specs PHOIBOS 150 MCD electron spectrometer (SPECS GmbH, Berlin, Germany) and X-ray tube with an Mg cathode (hν = 1253.6 eV) were used to obtain X-ray photoelectron spectra (XPS). The measurements were carried out under vacuum conditions (4 × 10^−8^ Pa). The spectra were obtained in the constant transmission energy mode. XPS background subtraction was performed in accordance with Shirley method. The spectra deconvolution was carried out according to the set of mixed Gauss/Lorentz peaks; Casa XPS 2.3.23 software (Casa Software Ltd., Teignmouth, UK) was used. The quantification of atomic content was performed by means of sensitivity factors (elemental library of CasaXPS).

### 2.3. Standard Contact Porosimetry

The method of standard contact porosimetry (MSCP) [31,32] was applied to the investigation of porous structure and hydrophilic–hydrophobic properties of rGO/TiO_2_ nanocomposite. This technique allows us to establish pore structure of any materials in the widest possible diapason of pore radii (from 1 nm to 300 μm), and also to research hydrophilicity–hydrophobicity. This is very important for supercapacitors, which operate in an aqueous electrolyte. The pore size distribution corresponds to all pores when octane is used as a working liquid. Only hydrophilic pores are determined with water. The MSCP has been recognized by IUPAC [33]. The phase composition and microstructural features of ceramics were investigated using X-ray diffraction and scanning electron microscopy in comparison with energy-dispersive methods [34].

### 2.4. Electrochemical Methods

First of all, an electrochemical cell was assembled. The cell included an electrode made of an rGO/TiO_2_ composite (4 mg) and counter electrode made of CH-900 AC cloth (200 mg), which was produced by Kuraray (Tokio, Japan). This allowed us to consider a change in the cell voltage as a change in potential of the working electrode. Preliminarily, the potential of carbon electrode was measured relatively standard reference electrode. This gave a possibility to coordinate the potential measured by carbon with the standard hydrogen potential. All the potentials are given relative to the stationary potential of the CH-900 counter electrode, which is practically polarized under experimental conditions. After assembly and preliminary cyclic polarization in the EDL region, of the double electric layer (EDL), the cell was left for 18 h for complete wetting of hydrophilic pores and stabilization. In some cases, CV curves are given as volt–farad dependences that are obtained with a method of dividing the currents by the potential scanning rate. The value, which corresponds to zero potential relative to the reversible hydrogen electrode (RHE), is shown as dotted curve for all potential scales. Further CV curves were measured in the EDL region (from −600 mV to 100 mV) with a scanning rate (*w*) of 10 mV/s. After this, an impedance spectrum was recorded in the field, which is close to stationary potential (this potential was stabilized after exposure of the newly assembled cell (frequency intervals are indicated in the figure captures)). Then eight serial CV curves were measured from the value of stationary potential (*w* = 10 mV/s), the measurements were carried out at −1400 mV to 150 mV.

Furthermore, the cell was kept at −1350 mV for 1 h to obtain maximal values of reversible hydrogen pseudocapacitance. After this, several successive CV curves were recorded (w = 10 mV/s) in the interval of −1400 mV to 150 mV until stabilization of the shape of the CVA curves.

When the maximum reversible pseudocapacitance was reached, the impedance spectra were measured in the cathodic region of the reversible pseudocapacitance (charged state). Then CV curves were recorded (w = 1 mV/s) in the interval of −1250 mV to −350 mV.

Furthermore, the dependence of the pseudocapacitance value on the time of cathode exposure (charging) was measured. The cathode charging was carried out under potentiostatic regime at the potential that is close to the cathodic minimum of current (according to the CV data at 1 mV/s). The charging time was varied from 5 s to 500 s. The current of anodic discharging was 1 mA. The value of specific pseudocapacitance per mass unit of TiO_2_ was calculated according to the data of galvanostatic curves.

Electrochemical experiments were performed using a P-40X potentiostat supplied with a FRA-24M module of electrochemical impedance (LLC “Elins”, Moscow, Russia).

As established according to the data of preliminary study of the electrochemical behavior of the rGO/TiO_2_ composite, this material shows an appearance of reversible maximum in the region of zero potentials relative to RHE. This phenomenon is observed in acidic electrolytes under deep cathodic polarization (electrochemical hydrogenation). The maximum value is controlled by the time of the electrode exposure in the region of negative potentials. However, in the case of strong acids (primarily sulfuric acid), the capacitance value of the maximum decreases quite rapidly over time. This decrease is independent of the electrode holding potential. Probably, this is caused by the gradual TiO_2_ dissolution in the acidic electrolyte. As a result of the experimental search, the purpose of which was to investigate the electrochemical behavior of the composite in acidic media, the electrolyte of optimal composition was chosen. This electrolyte was obtained by mixing equal volumes of 1 M orthophosphoric acid and 2 M zinc bromide solutions. For this highly conductive electrolyte, very slow decrease in the capacitance of investigated maximum was observed. This allows us to obtain reliable quantitative information about the process.

## 3. Results and Discussion

### 3.1. X-ray Photoelectron Spectra

Figure 2 shows a panoramic XPS spectrum of the nanocomposites containing 15 mass % TiO_2_. As seen, the peaks of carbon and oxygen are the most intensive. The surface composition of the sample, which was calculated from the integral intensities of peaks taking into account the corresponding photoionization cross-sections, is given in the insertion of Figure 2. As seen, the content of titanium is at the level of 0.08 at. %, although the synthesis method provides 15 mass % TiO_2_. This difference in bulk and surface titanium content means coating TiO_2_ particles with sheets of graphene oxide. Earlier we described this effect in [34]. The presence of nitrogen and sulfur in the analysis zone should be noted. A small amount of sulfur in a form of sulfogroups is due to the synthesis of graphite oxide according to the Hummers method [35].The presence of inconsiderable sulfur impurities in the GO samples was previously noted many times (see, for instance, [36]). The appearance of nitrogen is evidently caused by the reduction in GO samples with hydrazine vapor (see the Experimental Section and also [37]). Doping of rGO with nitrogen is considered beneficial from the point of view of increasing the capacitance of supercapacitors based on carbon materials [38,39,40].

Several versions of the description of the shape of C1s line are known. Here we will take into consideration the most commonly used version. The essence of this approach is that the positions of the C1s photoelectron peaks, which are due to the C sp^2^ and C sp^3^ states, are different from 0.4 eV [41] to 0.9 eV [42]. Thus, in order to determine the sp^2^/sp^3^ ratio, the C1s line has to be described by two or more peaks (depending on the experimental half-width of the line). As a rule, asymmetry of the peaks is not taken into account. Figure 3 gives one of the versions of this description. The positions of individual peaks, their half-widths and relative intensities are shown in Table 1. In accordance with the abovementioned literature data, the C1 and C2 peaks can be attributed to the C sp^2^ and C sp^3^ states, respectively. The peaks with higher bonding energies of carbon atoms have one bond with an oxygen atom (peak C3) and two bonds with an oxygen atom (peak C4) [43,44,45,46]. Both carbon atoms of carboxyl groups and losses due to plasmon excitation can make their contribution to the intensity of the C5 peak [39]. This description of the shape of C1s line gives a slightly overestimated content of C sp^3^ (12%–15%) [47].

Surface nitrogen-containing groups can be identified based on the analysis of the fine structure of the N1s line. According to the literature data [48] and the sources cited there, pyridine and pyrrol nitrogen appear in the ranges of 398.0–399.3 eV, and 399.8–401.2 eV, respectively. It should be noted that the nitrogen of amino- (399.1 eV) and cyano- (399.3 eV) groups gives peaks in this region. Therefore, it is difficult to identify it clearly. The peak of central graphite nitrogen is located about 401 eV, and the peak of terminal graphitic nitrogen is at 402.3 eV. Oxidized pyridine nitrogen is expressed as a peak at 402.8 eV. The peak at 404.7 eV corresponds to physically adsorbed N_2_ [48]. The N1s spectrum of the composite containing 15% TiO_2_ (Figure 3 N1s) can be divided into two peaks.

The assignment of the peaks allows us to conclude that nitrogen atoms of pyridine groups (peak N1) make the main contribution to the N1s spectrum. Both central nitrogen atoms of graphite and nitrogen atoms of pyrrole groups can form the N2 peak.

### 3.2. SEM

In Figure 4a, which shows a micrograph of the aerogel of the nanocomposite containing 30 mass % TiO_2_, one can see both transparent sheets of graphene oxide and rather large (up to several microns) aggregates of titanium dioxide. It can be seen that some TiO_2_ aggregates are located, as it were, in the folds of graphene oxide sheets. The other part of the units is attached to the GO sheets with only one side. We cannot yet say what the interface between the titanium dioxide aggregate and the GO sheet is. However, from the electrochemical data given below, it can be argued that this contact is rather short since titanium dioxide particles are effectively involved in electrochemical processes. Figure 3b shows a photograph of pure graphene oxide aerogel for comparison. It can be seen that there are no characteristic TiO_2_ inclusions in this photograph.

### 3.3. Porosimetric Data

The distributions of pore volume vs. effective radius (r*), which were measured with water and ideally wetting liquid (octane), are given in Figure 5a,b. The measurements were carried out for the rGO/TiO_2_ nanocomposite containing 30% TiO_2_. The wetting angle (*θ*) of octane is equal to 0° for all materials [49], the corresponding curve is related to all pores. The curve, which was obtained using water, is attributed only to hydrophilic pores. The distributions of pore volume and specific surface area (Figure 5c) are plotted vs. the effective pore radius (r*). This value is determined as r/cos *θ*. Here *r* is the true pore radius measured in octane, *θ* is the wetting angle of water. For the octane curve, r* = r, since cos *θ* = 1. Regarding the curve measured with water, cos*θ* < 1 and r* > r. As follows from Fig 5 and Table 2, both hydrophilic and hydrophobic pores are characteristic for the sample. Nevertheless, the fraction of hydrophobic pores is rather small. The total pore volume is 16.3 cm^3^/g, the volume of hydrophilic and hydrophobic pores is 13.3 cm^3^/g and 3 cm^3^/g, respectively. In accordance with calculations [31,32], the value of total SSA was estimated as 227 m^2^/g. Hydrophilic SSA reaches only 93 m^2^/g.

According to the differential pore size distributions (Figure 5b), these curves are characterized by two main maxima: they are located in the region of small (from 30 nm to 100 nm) and large (from 10^4^ nm to 10^5^ nm). Since rGO is a more highly dispersive material compared with c TiO_2_, it is possible to suppose that the maximum in the region of small r* values is related mainly to rGO. Large pores are probably attributed to TiO_2_. As follows from Fig 4c, pores with a radii of 2–200 nm make a contribution to the SSA. The main contribution is due to pores, the radius of which is 10–200 cm.

Figure 5d illustrates the distribution of the wetting angle vs. true pore radius. It is interesting that the pores of 30–100 nm are hydrophilic (*θ* = 0). An interesting fact follows from Figure 4 that pores with r from 30 nm to 100 nm are hydrophilic (*θ* = 0), while for pores with r < 30 nm and with r > 100 nm, the values of *θ* increase sharply.

This is due to the hydrophilicity of TiO_2_ on the one hand. On the other hand, both hydrophilic and hydrophobic pores are attributes for rGO. This is typical for carbon materials.

### 3.4. Electrochemical Measurements

Figure 6 illustrates cyclic volt–faradaic curves (CVF), which were obtained by successive cycling of electrodes from composites with different contents of TiO_2_. The CVF curves show successive growth of the reversible maximum, which is formed after the electrode polarization in the region of negative potentials. The first cycles for both curves show significant irreversibility of the cathode half cycle. This can be caused by electrochemical reduction in residual oxygen. Simultaneously with the increase in the capacitance of the reversible maximum, a slight but systematic decrease in the capacitance of the EDL is observed.

Moreover, anodic currents form a sloping maximum in the region of negative potentials. The maximum also tends to decrease as the reversible maximum grows. To reach the limiting value of the reversible maximum, the electrode was exposed at −1.3 V (relative to carbon) during 1 h. This allowed us the possibility to obtain stable CVF curves with a maximal capacitance value (Figure 7). The figure also shows the curves of initial EDL capacitance, which were obtained before deep cathodic polarization. The decrease in scanning rate causes a sharp decrease in the spread between the cathode and anode maxima. This demonstrates high reversibility of the processes, which determine these maxima. As shown from Figure 8, the average potential of the current-forming process is close to zero potential RHE. This can indicate surface or volume hydration–dehydration of TiO_2_.

In order to obtain quantitative information about the capacitance and kinetic characteristics of the observed reversible process, galvanostatic investigations of anodic discharging were performed under different times of potentiostatic charging. Figure 9 show families of anodic discharging curves of pseudocapacitance. The pseudocapacitance was obtained as a result of TiO_2_ hydrogenation in a wide range of exposure times at −1.2 V (carbon). The discharging current of 240 mA/g was chosen so that the discharge time did not exceed the maximum charging time of 500 s.

According to the data of Figure 9a (15% TiO_2_) and Figure 9b (30% TiO_2_), the dependencies of the values of specific charge (converted to the TiO_2_ mass) were plotted vs. the square root of charging time (Figure 10). As seen, the capacitance values come close to each other in the region of high magnitudes of charging time. This indicates that TiO_2_ determines the current-forming reaction. Moreover, the curves show a break at a characteristic time of about 50 s (this parameter determines the time of change in the mechanisms of kinetic control of the main reaction). Linearity of the curves indicates that cathodic charging is most likely controlled by diffusion below 50 s. Such a short time interval for diffusion control of a solid-phase reaction can indicate hydrogen, which is capable of reversible reduction/oxidation. For the conditions of hydrogenated TiO_2_ reduced in acid, hydrogen is present in the solid phase volume, capable of reversible reduction/oxidation. The total charge of the current-forming process is close to 700 C/g (TiO_2_). It is higher than ½ electrons per TiO_2_ formula unit. In other words (taking into account incomplete availability of TiO_2_ in the composite), the hydrogenated form can be considered as an interstitial compound with an empirical formula close to TiOOH.

Comparing Figure 9a (15% TiO_2_) and Figure 9b (30% TiO_2_) shows the capacitance of the composite increases with increasing TiO_2_ content. This is explained by the fact that the pseudocapacity of the composite lies in the TiO_2_ charge/discharge reaction. Overall, the capacities of our composites measured in an acidic electrolyte are somewhat lower than those obtained in an alkaline electrolyte [50]. However, in contrast to acidic electrolytes, the observed pseudocapacity of TiO_2_ in alkaline electrolytes is more efficiently used in the region of the operation of the positive electrode of supercapacitors.

Figure 11 shows impedance spectra, which are given in the form of a Nyquist diagram, for the regions of EDL charging and reversible reaction. The spectra of double-layer regions are characterized by a classical capacitive form with a distributed nature of charging. The complex shape of the spectra in the region of the reversible reaction is determined by the mutual influence of the simultaneous surface charging rOG and the fast Faraday process (the spectrum of which has the shape of a semicircle). In the region of low frequencies, the spectrum shape can be complicated by a self-discharge current. It increases the values of the real part of impedance. The main differences in the impedance of composites with various TiO_2_ is expressed in the region of low frequencies (EDL region). In this case, the real part of impedance is lower than that of the Faraday reaction for the sample containing 15% TiO_2_. This can be explained by a higher electric conductivity of the composite with lower TiO_2_ content.

More detailed comparative analysis can be obtained from the data, which are given as Bode diagram. Figure 12 illustrate these diagrams for the region of mid frequencies. The dotted f1 and f2 lines are attributed to characteristic frequencies for the imaginary part of impedance in the region of the faradaic process. Here f1 shows the maximum frequency of the beginning of the spectrum deviation from the total capacitive charging (inflection point of curve 3). f2 corresponds to the frequency at which the imaginary parts (capacitance) are equal for faradaic and double-layer charging. In general, f2 is the characteristic frequency, above which the capacitance measured in the region of EDL charging exceeds the capacitance measured in the region of reversible reaction. A comparison of the Bode diagrams of Figure 12a,b shows that an increase in the TiO_2_ content in the composite results in a shift of the characteristic frequencies towards higher values.

According to the results of impedance studies, a fitting was carried out in order to develop a simplified equivalent electrical circuit. The equivalent circuit and fitting data are shown in Figure 13. The most important elements of the resulting circuit are: R2 is the electron transfer resistance in the main current-generating reaction, C is an element showing the total reversible capacitance and W is an element that determines the diffusion control of charging.

As can be seen from the figure, the equivalent circuit describes the experimental data with high accuracy.

## 4. Conclusions

A highly porous rGO/TiO_2_ nanocomposite was synthesized, and its electrochemical properties were studied to evaluate the possibility to use it as an electrode for supercapacitors. The characterizations were carried out using such methods as XPS, impedance spectroscopy, CVA, etc. The peculiarity of this work in comparison with previous ones is that granular aerogel rGO/TiO_2_ was used as the starting material for the manufacture of the electrode. For the aerogel synthesis, industrial TiO_2_ Hombikat UV100 with a high specific surface area and anatase structure was used. The reduction of the aerogel was carried out with hydrazine vapor, which is associated with doping of graphene oxide hydrogenation in the region of zero potential on the scale of a standard hydrogen with nitrogen. An important feature of this work is the study using the MSCP method to estimate the porous structure and hydrophilic–hydrophobic properties of the nanocomposite. This is important for a supercapacitor, which involves an aqueous electrolyte.

The following results were obtained:

It was found that the hydrophilic surface area was approximately half of the total surface area of the nanocomposites.

As a result of the electrochemical electrode, a reversible Faraday reaction with a high recharging rate (exchange currents) was observed.

The characteristic charging time of the Faraday reaction does not exceed several tens of seconds, which makes it possible to consider the use of this pseudocapacitor in fast energy storage systems such as hybrid supercapacitors.

A sufficiently high limiting pseudocapacitance of the reaction was obtained at about 1200 C/g TiO_2_.

An assumption was made about the mechanism of the process: TiOOH ↔ TiO_2_(H^+^)_i_ + e^−^ (i–intercalated).

Since the hydrophilic surface area of the nanocomposite is only about half of its total area, the nanocomposite requires further hydrophilization in order to obtain higher electrochemical characteristics of the electrode.

## Figures and Tables

**Figure 1 materials-15-07856-f001:**
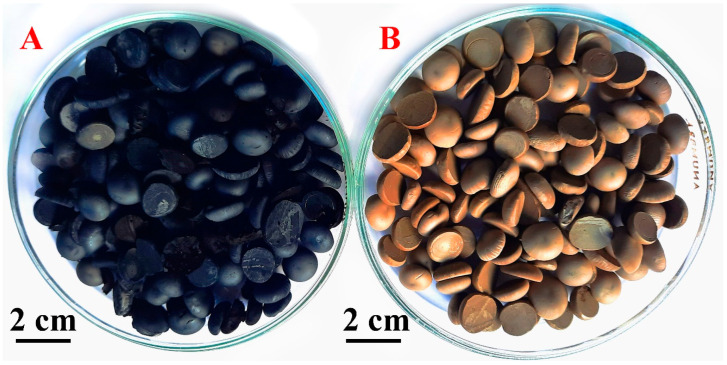
Photographs of composite aerogel after reduction (**A**) and before reduction (**B**).

**Figure 2 materials-15-07856-f002:**
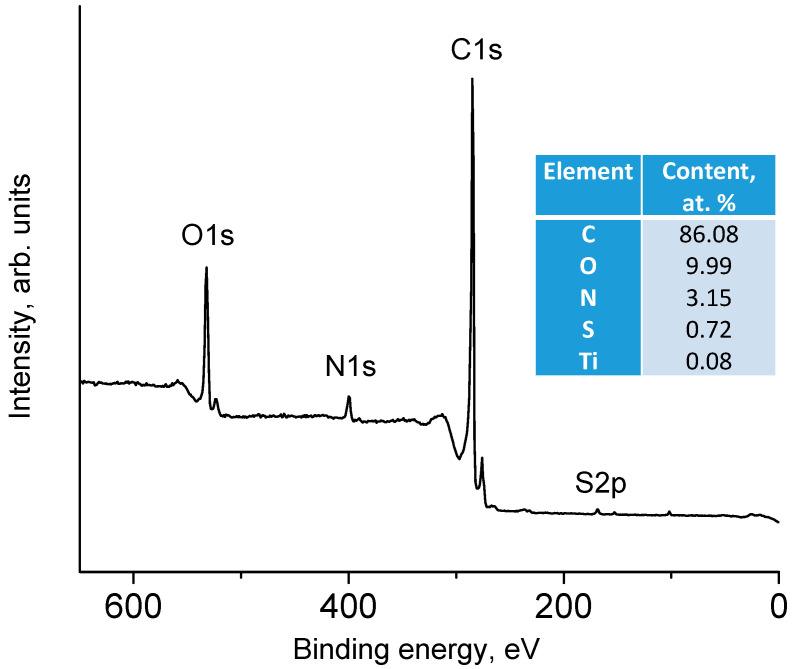
Survey XPS spectrum of the nanocomposites containing 15 mass % TiO_2_. Insertion: elemental composition of the sample surface.

**Figure 3 materials-15-07856-f003:**
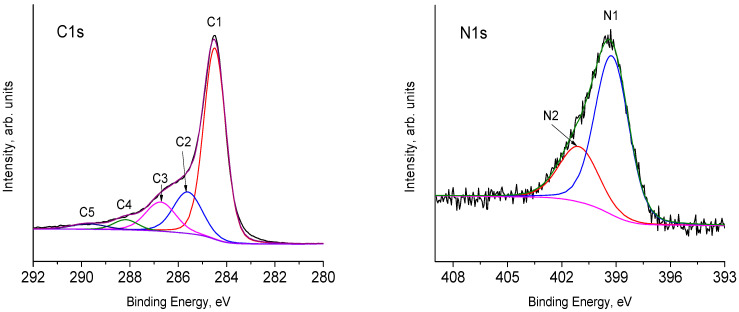
High-resolution XPS spectra of C1s and N1s of the nanocomposites containing 15 mass % TiO_2_.

**Figure 4 materials-15-07856-f004:**
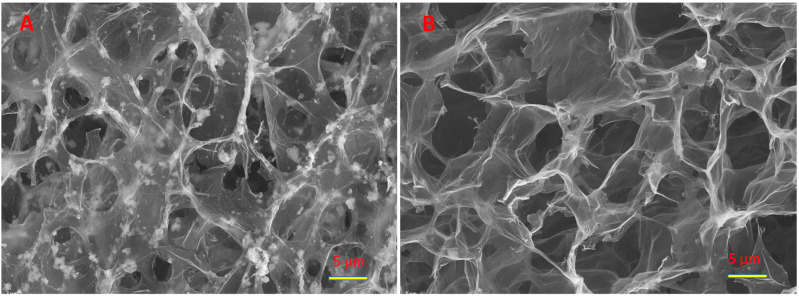
SEM micrographs of composite (**A**) and pure (**B**) aerogels.

**Figure 5 materials-15-07856-f005:**
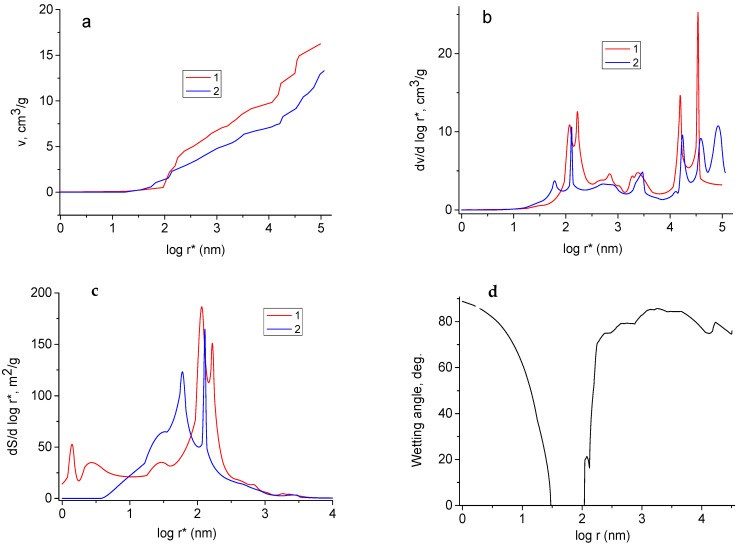
Results of porosimetric investigation for the rGO/TiO_2_ nanocomposite containing 30% TiO_2_: (**a**) integral pore volume distribution; (**b**) differential pore volume distribution; (**c**) differential pore surface distribution; (**d**) wetting angle as a function of pore radius. Data obtained using octane (1) and water (2) as working fluids.

**Figure 6 materials-15-07856-f006:**
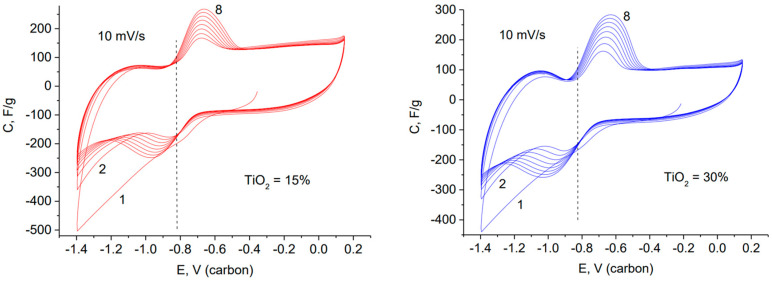
Cyclic cathodic polarization of the composite with 15 wt. % and 30 wt. % TiO_2_; 8 successive cycles are shown, they lead to an increase in the reversible hydrogen pseudocapacity.

**Figure 7 materials-15-07856-f007:**
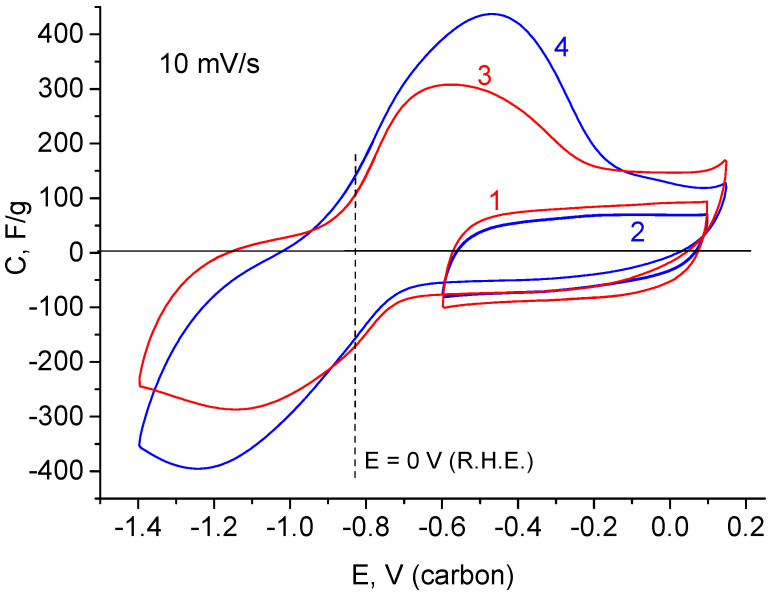
CVF curves: 1—rGO/TiO_2_ 15% (EDL region before cathodic polarization), 2—rGO/TiO_2_ 30% (EDL region before cathodic polarization), 3—rGO/TiO_2_ 15% (after deep cathodic polarization), 4—rGO/TiO_2_ 30% (after deep cathodic polarization).

**Figure 8 materials-15-07856-f008:**
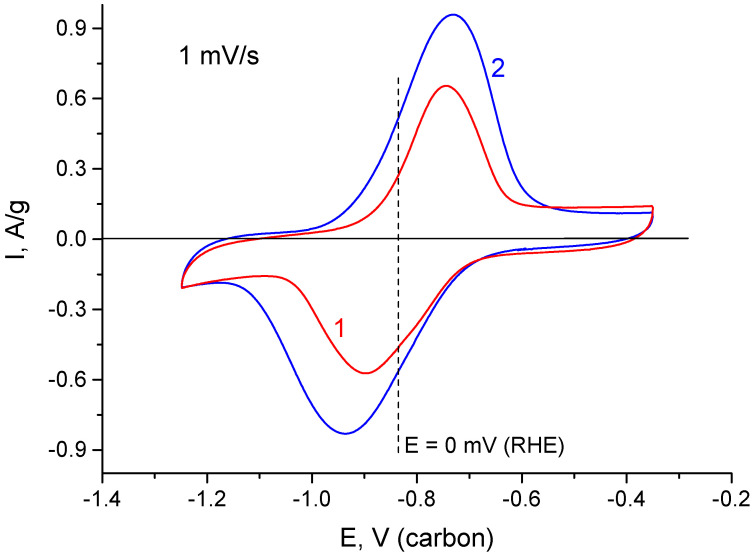
CVA curves (at 1 mV/s) in the field of reversible hydrogen pseudocapacitance after deep cathodic polarization: 1—rGO/TiO_2_ 15%; 2—rGO/TiO_2_ 30%.

**Figure 9 materials-15-07856-f009:**
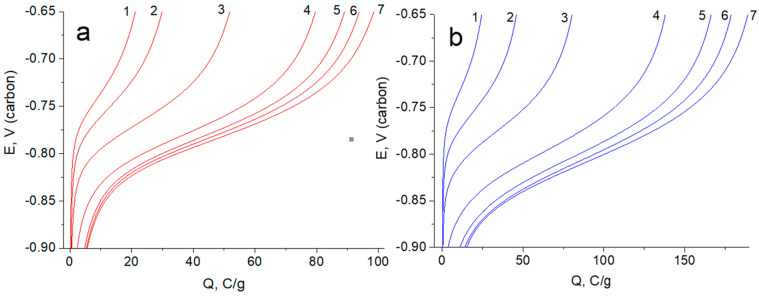
Galvanostatic charging curves of rGO/TiO_2_ 15% (**a**), rGO/TiO_2_ 30% (**b**) and composites, which were obtained at discharging current of 240 mA/g for different time of potentiostatic charging at −1.2 V: 1—5 s; 2—10 s; 3—20 s; 4—50 s; 5—100 s; 6—200 s; 7—500 s.

**Figure 10 materials-15-07856-f010:**
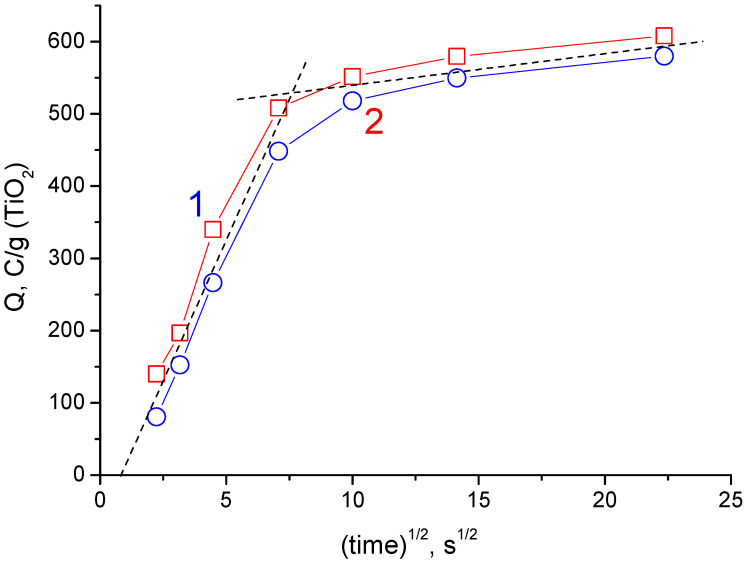
Dependence of pseudocapacitance, which was obtained from the data of galvanostatic discharging (see Figure 8) and recalculated per mass unit of TiO_2_, on the square root of charging time. 1—rGO/TiO_2_ 15%; 2—rGO/TiO_2_ 30%.

**Figure 11 materials-15-07856-f011:**
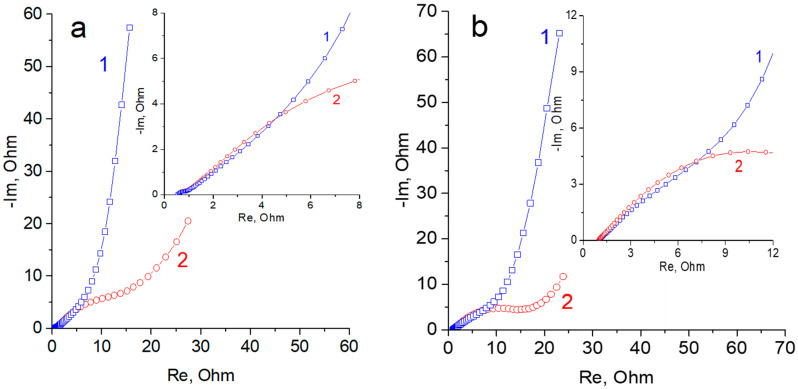
(**a**) Nyquist diagram for the rGO/TiO_2_ 15% composite for different states of the electrode: 1—EDL region (−150 mV); 2—region of charged pseudocapacitance of TiO_2_ (−880 mV). The insertion illustrates the region of mid frequencies in large scale. The frequency diapason was 100 kHz to 10 MHz. (**b**) Nyquist diagram for the rGO/TiO_2_ 30% composite for different states of the electrode: 1—EDL region (−200 mV); 2—region of charged pseudocapacitance of TiO_2_ (−900 mV). The insertion illustrates the region of mid frequencies in large scale. The frequency diapason was 100 kHz to 10 MHz.

**Figure 12 materials-15-07856-f012:**
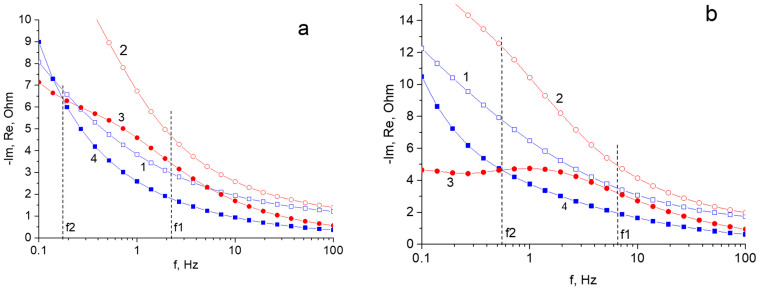
(**a**) Region of Bode diagram (from 100 Hz to 0.1 Hz) for the rGO/TiO_2_ 15% composite: 1—(Re) for the EDL region (−150 mV); 2—(Re) for the region of charged pseudocapacitance of TiO_2_ (−880 mV); 3—(−Im) for the region of charged pseudocapacitance of TiO_2_ (−880 mV); 4—(−Im) for the EDL region (−150 mV). (**b**) Region of Bode diagram (from 100 Hz to 0.1 Hz) for the composites of rGO/TiO_2_ 30%: 1—(Re) for the EDL region (−200 mV); 2—(Re) for the region of charged pseudocapacitance of TiO_2_ (−900 mV); 3—(−Im) for the region of charged pseudocapacitance of (−900 mV); 4—(−Im) for he EDL region (−200 mV).

**Figure 13 materials-15-07856-f013:**
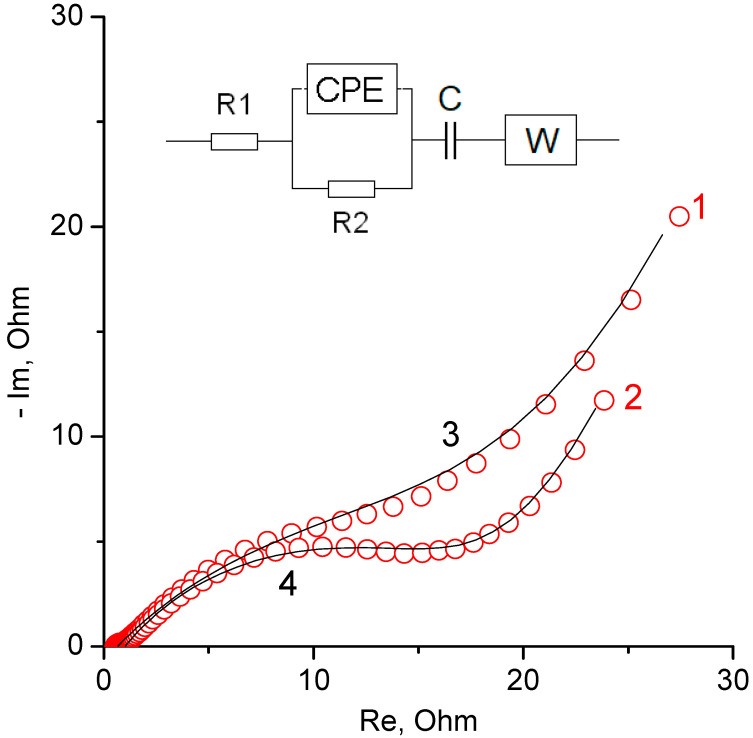
Equivalent electrical circuit and Nyquist diagram for experimental spectra (red circles) and for spectra obtained by fitting an equivalent electrical circuit (black lines). 1, 3—for rGO/TiO_2_ 15% composite; 2, 4—for rGO/TiO_2_ 30% composite.

**Table 1 materials-15-07856-t001:** Position and intensity of C1s peaks.

Peak	Position, eV	Content, %
C1	284.52	61.07
C2	285.31	12.21
C3	286.41	20.1
C4	288.13	4.13
C5	289.78	2.49

**Table 2 materials-15-07856-t002:** Main characteristics of porous structure of the rGO/TiO_2_ composite.

Hydrophilic Porosity cm^3^/g	Hydrophobic Porosity cm^3^/g	Total Porosity cm^3^/g	Hydrophilic Specific Surface Area m^2^/g	Total Specific Surface Area m^2^/g	Average Wetting Angle by Water
13.3	3.0	16.3	93	227	63.2

## Data Availability

Not applicable.

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
