# Peer review of "Supercapacitor Properties of rGO-TiO2 Nanocomposite in Two-component Acidic Electrolyte"

_materials, 2022, doi:10.3390/ma15217856_

Round 1
Reviewer 1 Report
Referee report on “Supercapacitor properties of rGO-TiO2 nanocomposite in two-component acidic electrolyte” by Yury M. Volfkovich et al
This is a quite interesting article that probably can be recommended for publication, but after major revision with clarifying and detailing some parts of the text.
1. Abstract. Abbreviation rGO (reduced graphene oxide) must be deciphered. This is important for a wider readership.
2. Introduction. 1 paragraph. First and second sentences need supporting references. Although, this paragraph contains important information, the relevance and novelty of which is in doubt, because only one rather old book of 2013 is confirmed. References for this paragraph should be revised.
3. The same for second paragraph of the Introduction. There are no supporting references at al.
4. 3rd paragraph. The same requirement.
5. 4th paragraph. Although quite a few oxides are listed here, references 3, 4 contain information only about RuO2 and NiO. No references for other oxides.
6. Further, the introduction contains a lot of chaotic information with many details and figures, the importance of which does not follow at all from the abstract. How important the second part of the fifth paragraph is is not clear.
7. 6th paragraph. First sentence has no verb.
8. The continuation of the introduction is too long, chaotic and the main line is not visible. It even seems that the authors simply collected information from the abstracts of other articles, not much bothering to highlight the main and important points.
9. Figure 1 needs to be scaled to see dimensions.
10. Paragraph “X-ray photoelectron spectra” Line 6 – grapheme oxide.
11. Fig. 4 also needs to be scaled to see dimensions.
12. It seems obvious that in this article the pores are quite complex and different. It is clear, that porosimetric measurements greatly simplify the picture. Other methods, such as positron annihilation spectroscopy, would be more informative here. See, for example: Klym, H.; Karbovnyk, I.; Luchechko, A.; et al . Evolution of Free Volumes in Polycrystalline BaGa2O4 Ceramics Doped with Eu3+ Ions. Crystals 2021, 11, 1515. https://doi.org/10.3390/cryst11121515
13. It would be useful to supplement the obtained data with measurements of raman and photoluminescence. Last measurements allow to see all structural anatase-to-rutile phase features: Serga, V., Burve, R., Krumina, A., et al (2021). Study of phase composition, photocatalytic activity, and photoluminescence of TiO2 with Eu additive produced by the extraction-pyrolytic method. journal of materials research and technology, 13, 2350-2360.
Understanding that there are multiple modifications is an absolute step towards understanding TiO2 composites:
Tsebriienko, T.; Popov, A.I. Effect of Poly(Titanium Oxide) on the Viscoelastic and Thermophysical Properties of Interpenetrating Polymer Networks. Crystals 2021, 11, 794. https://doi.org/10.3390/cryst11070794
All of the above should be reflected in the text, even if only as a discussion and analysis.
14. In the conclusions, it is necessary to clearly formulate what new data about the studied materials were obtained in this work?
15. The manuscript contains many typographical errors and requires a detailed reading from the native speaker.
In general, the manuscript is interesting and can be recommended for publication after constructive reflection on the above comments.
Author Response
Reply to Reviewer # 1 Comments
This is a quite interesting article that probably can be recommended for publication, but after major revision with clarifying and detailing some parts of the text.
- Abstract. Abbreviation rGO (reduced graphene oxide) must be deciphered. This is important for a wider readership. Authors reply: The remark is fair. The new version of the manuscript says "reduced graphene oxide/titanium dioxide (rGO/TiO2)".
- Introduction. 1 paragraph. First and second sentences need supporting references. Although, this paragraph contains important information, the relevance and novelty of which is in doubt, because only one rather old book of 2013 is confirmed. References for this paragraph should be revised. Authors reply: References [3-5] have been added to the new version of the manuscript.
- The same for second paragraph of the Introduction. There are no supporting references at al. Authors reply: References [6-8] have been added to the new version of the manuscript.
- 3rdparagraph. The same requirement. Authors reply: References [9-12] have been added to the new version of the manuscript.
- 4thparagraph. Although quite a few oxides are listed here, references 3, 4 contain information only about RuO2 and NiO. No references for other oxides. Authors reply: The new version of the manuscript contains links to reviews on metal oxide-based supercapacitors
- Further, the introduction contains a lot of chaotic information with many details and figures, the importance of which does not follow at all from the abstract. How important the second part of the fifth paragraph is is not clear. Authors reply: The second part of the fifth paragraph has been deleted.
- 6thparagraph. First sentence has no verb. Authors reply: This error has been corrected in the new version of the manuscript.
- The continuation of the introduction is too long, chaotic and the main line is not visible. It even seems that the authors simply collected information from the abstracts of other articles, not much bothering to highlight the main and important points. Authors reply: The introduction has been drastically shortened. Links are left only to important studies for this topic.
- Figure 1 needs to be scaled to see dimensions. Authors reply: In this figure, the scale can be determined by the size of the petri dish (100 mm). Nevertheless, in accordance with the recommendation of the Reviewer, we indicated the scale directly in the figure.
- 1Paragraph “X-ray photoelectron spectra” Line 6 – grapheme oxide. Authors reply: The remark is fair. The new version of the manuscript says "graphene oxide".
- Fig. 4 also needs to be scaled to see dimensions. Authors reply: A strange remark, in Figure 4 the scale is indicated.
- It seems obvious that in this article the pores are quite complex and different. It is clear, that porosimetric measurements greatly simplify the picture. Other methods, such as positron annihilation spectroscopy, would be more informative here. See, for example: Klym, H.; Karbovnyk, I.; Luchechko, A.; et al . Evolution of Free Volumes in Polycrystalline BaGa2O4Ceramics Doped with Eu3+ Ions. Crystals 2021, 11, 1515. https://doi.org/10.3390/cryst11121515 Authors reply: Thanks to the reviewer for information about the interesting work. In the new version of the manuscript, we cited it as a reference [34].
- It would be useful to supplement the obtained data with measurements of raman and photoluminescence. Last measurements allow to see all structural anatase-to-rutile phase features: Serga, V., Burve, R., Krumina, A., et al (2021). Study of phase composition, photocatalytic activity, and photoluminescence of TiO2 with Eu additive produced by the extraction-pyrolytic method. journal of materials research and technology, 13, 2350-2360
Understanding that there are multiple modifications is an absolute step towards understanding TiO2 composites:
Tsebriienko, T.; Popov, A.I. Effect of Poly(Titanium Oxide) on the Viscoelastic and Thermophysical Properties of Interpenetrating Polymer Networks. Crystals 2021, 11, 794. https://doi.org/10.3390/cryst11070794
All of the above should be reflected in the text, even if only as a discussion and analysis. Authors reply: In this work, commercially available anatase titanium dioxide was used as the starting material (below, a certificate and Raman spectrum are attached especially for the reviewer). We did not carry out any treatments that could change the phase composition of TiO2 during the preparation of aerogels. By the way, some of the authors of this work have previously studied phase transitions in nanosized titanium dioxide samples (see Y. M. Shulga, E. N. Kabachkov, D. V. Matyushenko, E. N. Kurkin, and I. A. Domashnev “Thermally Stimulated Transformations in Brookite Containing TiO2 Nanopowders Produced by the Hydrolysis of TiCl4”, Technical Physics, 2011, Vol. 56, No. 1, pp. 97–101). Based on this experience, one could expect the appearance of a rutile phase upon heating to 440-500 oC. During the preparation of the studied composites, heating did not exceed 60 oC.
- In the conclusions, it is necessary to clearly formulate what new data about the studied materials were obtained in this work? Authors reply: In the new version of the manuscript, we partially reformulated the conclusions.
- The manuscript contains many typographical errors and requires a detailed reading from the native speaker Authors reply: This work has been done
In general, the manuscript is interesting and can be recommended for publication after constructive reflection on the above comments.
Thanks to the Reviewer for helpful comments!
Raman spectrum of Hombikat UV100

Reviewer 2 Report
The paper is presenting the synthesis and characterization of highly porous rGO/TiO2 nanocomposite. Besides, the hydrophilic and hydrophobic properties of the nanocomposite were studied. The subject of this study might be interesting, although there are some flaws that should be considered as follow:
1. There are some extra spaces in the text such as the first line of the experimental section, extra full stops such as the first line of the standard contact porosimetry section. Figure 1 caption mentioned "airgel" which I think the author meant aerogel and etc.
2. The introduction part has too much content. I believe a more concise and to-the-point introduction would help the reader to understand better. I think the authors should give the important aspect of references reviewed to present their work.
3. The novelty and importance of the study should be highlighted.
4. Please describe why you have used XPS?! I mean how does the XPS result help the concept?! As widely known, the XRD method can be used to understand the phase evolution, and Raman spectroscopy to define Id and Ig to show that rGO is formed. You can use these references for comparison:
a) https://doi.org/10.1007/s10854-020-04062-7
5. HRTEM or at least FESEM might be helpful to understand the microstructure of formed aerogel.
6. In the results the authors mentioned nanocomposites containing different mass % TiO2 (15 and 30) while there is no description in the experimental section. Please describe the synthesis method more precisely.
7. In the cyclic volt-faradaic curves, please define the corresponding reaction with the peaks in cyclic cathodic polarization!
8. Please provide the galvanostatic discharging curves as well for better understanding! Besides, please compare your result with others for example http://dx.doi.org/10.22068/ijmse.17.4.55
9. Please provide an equivalent circuit for EIS data
10. In the conclusion, please give some reason for the behavior resulted from your characterization. In this format, the study is more like a technical report rather than a scientific research paper!
11. I suggest that the authors give more reasoning for all presented data. All the characterizations were presented quite enough but the reasoning for this behavior and why there is a difference in the samples by changing TiO2 content is not presented!
Author Response
Reply to Reviewer # 2 Comments
The paper is presenting the synthesis and characterization of highly porous rGO/TiO2 nanocomposite. Besides, the hydrophilic and hydrophobic properties of the nanocomposite were studied. The subject of this study might be interesting, although there are some flaws that should be considered as follow:
- There are some extra spaces in the text such as the first line of the experimental section, extra full stops such as the first line of the standard contact porosimetry section. Figure 1 caption mentioned "airgel" which I think the author meant aerogel and etc. Authors reply: The remark is fair. These deficiencies have been corrected.
- The introduction part has too much content. I believe a more concise and to-the-point introduction would help the reader to understand better. I think the authors should give the important aspect of references reviewed to present their work. Authors reply: The remark is fair. In the new version of the manuscript, the volume of the introduction has been significantly reduced.
- 3. The novelty and importance of the study should be highlighted. Authors reply: In the abstract of the new version of the manuscript it is written: “Granular aerogel rGO/TiO2 was used as an initial material for the electrode manufacture at first time.”
- Please describe why you have used XPS?! I mean how does the XPS result help the concept?! As widely known, the XRD method can be used to understand the phase evolution, and Raman spectroscopy to define Id and Ig to show that rGO is formed. You can use these references for comparison: а) https://doi.org/10.1007/s10854-020-04062-7 Authors reply: The XPS method indicates that the performed treatments lead not only to reduction, but also to doping of rGO with nitrogen. The latter cannot be convincingly proved either by XRD or by Raman spectroscopy. In this work, commercially available anatase titanium dioxide was used as the starting material (below, in the end of our comments, Raman spectrum is attached especially for the reviewer). We did not carry out any treatments that could change the phase composition of TiO2 during the preparation of aerogels. By the way, some of the authors of this work have previously studied phase transitions in nanosized titanium dioxide samples (see Y. M. Shulga, E. N. Kabachkov, D. V. Matyushenko, E. N. Kurkin, and I. A. Domashnev “Thermally Stimulated Transformations in Brookite Containing TiO2 Nanopowders Produced by the Hydrolysis of TiCl4”, Technical Physics, 2011, Vol. 56, No. 1, pp. 97–101). Based on this experience, one could expect the appearance of a rutile phase upon heating to 440-500 oC. During the preparation of the studied composites, heating did not exceed 60 oC.
- HRTEM or at least FESEM might be helpful to understand the microstructure of formed aerogel. Authors reply: It is not clear why our SEM micrographs of composite rGO/TiO2 (A) and pure GO (B) aerogels (Fig. 4) were not noticed by the Reviewer.
- In the results the authors mentioned nanocomposites containing different mass % TiO2 (15 and 30) while there is no description in the experimental section. Please describe the synthesis method more precisely. Authors reply: The remark is fair. Appropriate clarifications have been made in the new version of the manuscript (see the experimental part).
- In the cyclic volt-faradaic curves, please define the corresponding reaction with the peaks in cyclic cathodic polarization! Authors reply: Most works on the electrochemical behavior of composites based on titanium oxide are limited to conclusions showing the numerical values of the specific capacitance of these materials, without conducting a deep analysis of the mechanism of current-forming reactions. In our case, when discussing the results, a number of features were shown that indirectly indicate the mechanism of the current-forming reaction. At the request of the reviewer, we introduce the following sentence into the main conclusions of the work: Based on the obtained electrochemical data, an assumption was made about the mechanism of the main current-forming reaction TiO2 + H+ + e- ↔ TiOOH.
- Please provide the galvanostatic discharging curves as well for better understanding! Besides, please compare your result with others for examplehttp://dx.doi.org/10.22068/ijmse.17.4.558. Authors reply: Discharge galvanostatic curves are shown in Fig.9. Only in contrast to the article indicated by the reviewer, in which the electrolyte is alkali, we used an acidic electrolyte, so this composite is more effectively considered as the negative electrode of a supercapacitor. The discharge, in this case, is directed from the region of negative potentials towards positive potentials. Therefore, at the request of the reviewer, the following paragraph has been introduced into the new version of the manuscript:
Comparison Fig.9a (15% TiO2) and Fig.9b (30% TiO2) shows the capacitance of the composite increases with increasing TiO2 content. This is explained by the fact that the pseudocapacity of the composite lies in the TiO2 charge/discharge reaction. Overall, the capacities of our composites measured in an acidic electrolyte are somewhat lower than those obtained in an alkaline electrolyte [50]. However, in contrast to acidic electrolytes, the observed pseudocapacity of TiO2 in alkaline electrolytes is more efficiently used in the region of operation of the positive electrode of supercapacitors.
- Please provide an equivalent circuit for EIS data Authors reply: At the request of the reviewer, we fitted the imedansometry data and proposed an equivalent electrical circuit, which made it possible to describe the experimental data with high accuracy. A figure with an equivalent circuit and a fitting of experimental data has been added to the article. Additions have also been made to the text of new version of manuscript: “According to the results of impedance studies, a fitting was carried out in order to develop a simplified equivalent electrical circuit. The equivalent circuit and fitting data are shown in Fig. 13. The most important elements of the resulting circuit are: R2 is the electron transfer resistance in the main current-generating reaction, C is an element showing the total reversible capacitance and W is an element that determines the diffusion control of charging.”
- In the conclusion, please give some reason for the behavior resulted from your characterization. In this format, the study is more like a technical report rather than a scientific research paper! Authors reply: The manuscript says that we associate the features of the electrochemical behavior of our samples with the original method of manufacturing the electrode (for the manufacture of the electrode, we used the rGO/TiO2 nanocomposite aerogel, which, in turn, was made by freeze drying from a mixture of single-layer sheets of graphene oxide and titanium dioxide with a high specific surface and structure of anatase).
- I suggest that the authors give more reasoning for all presented data. All the characterizations were presented quite enough but the reasoning for this behavior and why there is a difference in the samples by changing TiO2 content is not presented! Authors reply: As a result of scientific research and comparison of the obtained data with the literature data, it was found that the capacitance of the electrodes increases with an increase in the content of TiO2. This is because the reason for the pseudo capacitance is the TiO2 charge/discharge reaction. (The corresponding conclusion has been inserted in the text).
Thanks to the Reviewer for helpful comments!
Raman spectrum of Hombikat UV100

Round 2
Reviewer 1 Report
The authors have replied: " 12. It seems obvious that in this article the pores are quite complex and different. It is clear, that porosimetric measurements greatly simplify the picture. Other methods, such as positron annihilation spectroscopy, would be more informative here. See, for example: Klym, H.; Karbovnyk, I.; Luchechko, A.; et al . Evolution of Free Volumes in Polycrystalline BaGa2O4Ceramics Doped with Eu3+ Ions. Crystals 2021, 11, 1515. https://doi.org/10.3390/cryst11121515 Authors reply: Thanks to the reviewer for information about the interesting work. In the new version of the manuscript, we cited it as a reference [34] "
I did not see the proposed changes in the new version of the manuscript.
Reviewer 2 Report
The revised version addressed all the raised issues and the paper can be accepted in its current form in the journal of Materials